# PD Flexible Built-In High-Sensitivity Elliptical Monopole Antenna Sensor

**DOI:** 10.3390/s22134982

**Published:** 2022-07-01

**Authors:** Hanting Zhang, Guozhi Zhang, Xiaoxing Zhang, Hanlu Tian, Changyue Lu, Jianben Liu, Yin Zhang

**Affiliations:** 1Hubei Engineering Research Center for Safety Monitoring of New Energy and Power Grid Equipment, Hubei University of Technology, Wuhan 430068, China; 1910201408@hbut.edu.cn (H.Z.); xiaoxing.zhang@outlook.com (X.Z.); 102110326@hbut.edu.cn (H.T.); 102100219@hbut.edu.cn (C.L.); yinz@hbut.edu.cn (Y.Z.); 2State Key Laboratory of Power Grid Environmental Protection, China Electric Power Research Institute, Wuhan 430074, China; liujianben@epri.sgcc.com.cn

**Keywords:** Gas Insulated Switchgear, partial discharge, flexible antenna, ultra-high frequency, monopole antenna

## Abstract

In view of the insufficient signal detection sensitivity of Gas Insulated Switchgear (GIS), partial discharge (PD), ultra-high frequency (UHF), and failure to conform with GIS surface structure when the existing rigid stereo structure UHF sensor is built in, this paper, using rectangular patch antenna equivalent technique, trapezoidal ground plane technique, and coplanar waveguide (CPW) feed line index asymptotic linearization technique, conducts research on a flexible built-in high-sensitivity elliptic monopole antenna. The flexible antenna, with a thickness of only 0.28 mm, can be kept at a voltage standing wave ratio (VSWR) of less than three in the 300 MHz to 3 GHz band under the curvature radius of 0, 100, 300, and 500 mm, and at less than two in the 650 MHz to 3 GHz band. Through the true 220 kV-GIS partial discharge experimental platform built to analyze the high frequency electromagnetic wave detection performance of the built-in flexible antenna, it is shown that the flexible built-in high-sensitivity elliptical monopole antenna designed in this paper can effectively detect the characteristic signals of high-frequency electromagnetic waves emitted by partial discharges with an average discharge amount below 10 pC.

## 1. Introduction

Gas Insulated Switchgear is widely used in power systems of various voltage levels because of its compact structure, small package, sound insulation performance, high reliability, and long maintenance cycle. Depending on the analysis report of the actual operation dynamics of the power grid, the main problem affecting the stable operation is the insulation fault, and partial discharge is one of the most important causes of insulation fault [1]. PD process will generate electrical pulses, ultrasonic waves, electromagnetic radiation, etc. Pulse current detection method, ultrasonic wave detection method [2], ultra-high frequency detection method [3], and other PD detection methods [4] are designed correspondingly. They work with the detection of SF_6_ and other gas partial discharge decomposition components of the sensor to determine the overall level of PD and determine the cause of PD [5,6]. UHF method detects the radiation of high-frequency electromagnetic waves in the PD process in the 300 MHz~3 GHz bands using UHF antenna sensors, which can effectively avoid the corona interference signal generated by the power site below the 200 MHz band. It has keen sensitivity and anti-interference ability, so the new GIS of 220 kV and above is required to install UHF monitoring system or reserve UHF monitoring interface.

According to the different installation locations, UHF sensors can be divided into two types: external and internal. The external UHF sensor detects the PD signal of GIS equipment for monitoring the electromagnetic wave signal leaked from the pot insulator casting hole, which has the advantage of simple installation and does not affect the safe operation of GIS equipment. At present, the common external sensors are magnetic antenna, microstrip antenna, and other sensors [7,8]; however, due to the serious attenuation of electromagnetic waves leaked through the casting hole and the influence of external corona interference, the external UHF method is less sensitive and has poor anti-interference characteristics. Although the internal UHF sensor needs to be installed inside the GIS in advance when the GIS leaves the factory, it is more sensitive because the detected electromagnetic wave signal is not attenuated through other places such as the GIS basin insulator casting hole, and its anti-interference ability is also stronger because the GIS is made with a metal shell, which can effectively shield the external corona interference. The most common internal sensors are mainly spiral antenna, monopole antenna, and other sensors [9,10]. Due to their simple structure and good radiation characteristics, monopole antennas are widely used in the field of UHF partial discharge sensors. In [11], a planar monopole antenna based on biomimetic geometry of Inga Marginata leaves was designed to solve the problems of optimal size, gain, radiation pattern, and sensitivity for PD detection. Reference [12] designs a compact UHF antenna based on an octagonal monopole antenna, which has the advantages of simple structure, good gain, and strong pulse processing capability. However, most of the existing internal UHF sensors use a rigid base, and the longitudinal dimension is too large, thus creating the problem that the installation cannot use a metal cover with GIS, and requires the complete barrel-type structure of the GIS equipment itself to be modified with a convex complex structure to avoid the potential risk of damaging the electric field distribution inside the equipment. Reference [13] designed a planar monopole antenna based on meandering technique technology, which can effectively detect PD signals after bending and deforming, and has a high signal-to-noise ratio. However, there are problems of low sensitivity and poor capture of UHF signals.

In response to the above problems, our team proposed the research of flexible internal UHF antenna sensor and developed different types of flexible Archimedean spiral antenna sensors. However, the flexible UHF antenna sensors are not effective in low frequency bands due to the structural principle of the spiral antenna itself.

At present, the common methods to improve the gain of monopole antenna include microstrip feeder taper, ground plane trapezoid, antenna double-layer polarization, meandering technology, antenna ellipse, and other methods [14,15,16]. Based on this, this paper proposes a study of PD detection of flexible internal high-sensitivity elliptical monopole antenna sensor, using rectangular patch antenna equivalent technique, trapezoidal ground plane technique, and CPW feed line index asymptotic linearization technique to construct a three-dimensional electromagnetic simulation model of flexible elliptical monopole antenna using ANSYSS HFSS software. The simulation obtains the characteristic parameters of VSWR and gain under different bending degrees of antenna, and develops the flexible antenna sensor sample according to the simulation optimization results, finally building a 220 kV GIS PD internal flexible antenna sensor testing test platform to experimentally test the developed flexible antenna sensor PD detection sensitivity.

## 2. Elliptical Radiation Patch Principle

The antenna part consists of copper elliptical radiation patch, trapezoidal ground plane, and CPW feed line, which are laid on the dielectric board.

### 2.1. Elliptical Plane Monopole Antenn

As a special type of micro strip antenna, elliptical planar monopole antenna has the advantages of simple structure, compact and wide band [17], and a feed network which can be integrated with the antenna body structure [18], which is why it is widely used in radar systems, military, and aerospace fields.

Elliptical radiation patch principle is shown in Figure 1a, where the long axis *a* and short axis *b* are determined by the maximum wavelength of electromagnetic waves it senses, and its specific size can be calculated according to the rectangular patch antenna equivalent theory, that is, as shown in Figure 1b, the bottom radius of *r* and the height of *h* cylindrical oscillator equivalent.

The area of the elliptical radiation patch S_s_ and the long axis of the ellipse 2*a* is equated with the column surface area S_c_ and height *h* of the cylindrical oscillator, respectively, i.e.,
(1)πab=2πrh
(2)2a=h
where *h* and the maximum wavelength of electromagnetic waves in the frequency band *λ* have the relations that:(3)$$h=\frac{0.24\lambda h}{r}(1+\frac{h}{r})$$

Through Equations (1)–(3), it can be concluded that the relationship between the lowest operating frequency *f* corresponding to the maximum wavelength of the electromagnetic wave sensed by the cylindrical oscillator and the long axis *a* and short axis *b* of the elliptical radiation patch is:(4)$$f=\frac{28.2}{8b+a}$$

When the elliptical radiation patch axis ratio (the elliptical radiation patch axis ratio can be expressed as AR = *a*/*b*) for 1, the elliptical radiation patch can be seen as a special circular radiation patch, so that the lowest working frequency and the relationship between the elliptical radiation patch size can be expressed as follows:(5)$$f=\frac{3.2}{b}$$

### 2.2. Trapezoidal Ground Plane

The trapezoidal ground plane adopts the idea of polarization of the discone antenna [19], taking the two-dimensional plane structure of the discone antenna as the ground plane of the antenna, so that the ground plane inherits the advantages of wide bandwidth, vertical line polarization, and omnidirectional radiation in the horizontal plane of the discone antenna [20]. Because the working band of the discone antenna is less than two in the UHF band, the width of the VSWR is eight octave bandwidth (octave bandwidth BW is expressed by Equation (6)), which can effectively fit the wide band range of PD UHF signal 300 MHz~3 GHz.
(6)$$BW=\frac{f_{H}}{f_{L}}$$

The directional characteristics of the discone antenna *ψ* (*γ,θ*) are calculated as
(7)$$\psi (\gamma ,\theta )=\sum rT_{v}(\theta )Z_{v}(k_{0\gamma })$$
where *T_v_* (*θ*) denotes a linear combination of *v*-order first and second-class Lejeune functions; *Z_v_* (*k_0γ_*) denotes the generalized *v*-order spherical Bessel function; and $k_{0}=\omega {\left(\varepsilon_{0}\mu_{0}\right)}^{1/2}$ denotes the free-space wave number.

Combining the spherical Bessel function and the Lejeune function, the functions of the inner and outer domains outside the vertebral body are taken as:

Inner domains:(8)$$A_{1}=\sum rT_{v}(\theta )J_{v}(k_{0\gamma })$$

Outer domains:(9)$$A_{2}=\sum nB_{n}P_{n}(\cos\theta )H_{n}(k_{0\gamma })$$

Among them:(10)$$T_{v}(\theta )=CvP_{v}(\theta )+D_{v}P_{v}(-\cos\theta )$$
(11)$$J_{v}(k_{0\gamma })=k_{0\gamma }j_{v}(k_{0\gamma })$$
(12)$$H_{n}(k_{o\gamma })=k_{0\gamma }h_{n}(k_{0\gamma })$$
where *P_n_* (cos*θ*) denotes the linear solution of the Legendre equation; *j_v_* (*k*_0_*_γ_*) denotes a standing wave of the v-order first class spherical Bessel function in the direction of *γ*; and *h_n_* (*k*_0_*_γ_*) denotes the outward traveling wave of the n-order second class spherical Hankel function along the direction of *γ*. *N* is a nonzero constant, *v* is a constant to be determined, and *B_n_*, *C_v_*, and *D_v_* are the expansion coefficients to be determined. Bringing in the boundary conditions, the directional map function of the discone antenna can be solved as:(13)$$f(\theta )=\sum nj_{n}-B_{n}(\cos\theta )P_{n}(\cos\theta )\sin\theta$$

The direction of antenna receiving electromagnetic wave concentration can be calculated by Equation (13).

The principle of the discone antenna is shown in Figure 2, the metal dish and the coaxial feed line of the vertebrae are connected, the diameter of the dish is *D*, the diameters of the upper and lower surface of the vertebrae are D_max_ and D_min_, the inclination angle of the vertebrae is *θ*_0_, and the bus length is *L*.

Because the discone antenna is equivalent to a half-tension angle of 90° and a half-tension angle of *θ*_0_ of the vertebrae composed of biconical antenna, its characteristic impedance is equivalent to half of the biconical antenna, which can be approximated by the Equation (14):(14)$$Z_{0}=60\ln(\cot\frac{\theta_{0}}{2})$$

In this paper, we adopt the plagiarization of the vertebral part of the discone antenna as the ground plane, to improve the overall impedance bandwidth of the antenna. The matching impedance of the feed line connected with the disk cone antenna is 50 Ω, so the size of the antenna can be designed according to the matching impedance during the design. The vertebral cone angle of the antenna cannot be too small, otherwise it will cause the matching impedance of the antenna to change drastically with the change of the bus *L*; therefore, the cone angle *θ*_0_ should be between 30~60°. The vertebral bus *L* cannot be too small, to avoid the problem that the reactance component becomes bigger due to the small radiation resistance of the antenna, which makes the matching impedance of the coaxial feed line become poor, so the vertebral bus *L* should be slightly larger than ¼ of the wavelength corresponding to the lowest working frequency of the antenna *λ*_max_, i.e.,
(15)$$L=k\times \lambda_{\max}/4$$
where the *k* values range from 1.3 to 1.5.

### 2.3. CPW Feed Line

CPW feed line, as a microwave planar transmission line with the advantages of low radiation loss, wide operating band, and simple structure, is often used in transmission lines and microwave millimeter wave integrated components and other aspects. Due to its unique planar structure, it can also be used in the antenna feeds, converting electromagnetic wave energy in space into electromagnetic signal [21].

The principle of CPW feed line is shown in Figure 3, where the middle guide tape and the ground plane on both sides are laid on the surface of the dielectric plate, the width of the guide tape is w, the distance between the guide tape and the ground plane is d_1_, and the thickness of the dielectric plate is h_1_.

CPW feed line characteristic impedance *Z*_01_ is calculated as:(16)$$Z_{01}=\frac{30\pi K^{\prime }(k)}{\sqrt{\varepsilon_{re}}K(k)}$$
where $K^{\prime }\left(k\right)=K(k^{\prime })$, $\;K^{\prime }\left(k\right)=K(k^{\prime })$, k=c/(c+2w); *K′ (k)* denotes the first class of complete elliptic cosine functions; and *K(k)* denotes the first class of complete elliptic functions.

When 0≤k≤0.7
(17)$$\frac{K(k)}{K^{\prime }(k)}={\left[\left(\frac{1}{\pi }\ln(2\frac{1+\sqrt{k^{\prime }}}{1-\sqrt{k^{\prime }}}\right)\right]}^{-1}$$

When 0.7≤k≤1
(18)$$\frac{K(k)}{K^{\prime }(k)}=\frac{1}{\pi }\ln(2\frac{1+\sqrt{k}}{1-\sqrt{k}})$$
where *ε_re_* denotes the effective dielectric constant of the CPW feed line, which is calculated as:(19)$$\varepsilon_{re}=\frac{\varepsilon_{r}+1}{2}\{\tan\left[1.75+0.775\ln(h_{1}/d_{1})\right]+\frac{k_{1}d_{1}}{h_{1}}\left[0.04-0.7k_{1}+0.01(0.25+k)(1-0.1\varepsilon_{r})\right]\}$$

After the above calculation, it can be obtained that the thickness of the dielectric plate has more influence on the characteristic impedance of CPW feed line. The antenna using CPW feed line technology can achieve the purpose of increasing the bandwidth of the antenna and also avoid the complex structural modification of the antenna due to the drastic fluctuation of the characteristic impedance.

## 3. Flexible High-Sensitivity Elliptical Monopole Antenna

The flexible high-sensitivity elliptical monopole antenna designed in this paper operates in the frequency band of 300 MHz~3 GHz, which can cover almost the entire UHF band. The antenna is designed for the lowest operating frequency of the operating band, and 1 GHz is taken as the center frequency of the antenna. From the above formula, the long axis a = 90 mm and the short axis b = 30 mm of the elliptical radiation patch before antenna optimization. The cone angle *θ*_0_ is taken as 30°, the bus length L = 175 mm of the cone ground plane, the upper base D_min_ = 3 mm, the lower base D_max_ = 180 mm, and the height H = 150 mm. The width w of CPW feed line is taken as 2.5 mm, the distance d_1_ = 0.1 mm between the guide tape and the ground plane, and the height H_1_ = 152 mm. The thickness of the antenna h_1_ is 0.28 mm. At this time, the antenna transverse length A = 180 mm and longitudinal length B = 210 mm are larger and need to be further reduced. In this paper, the exponential asymptotic impedance converter technique is used to realize the wide band of the antenna by exponential asymptotic linearization of the CPW feed line [22], and the designed parameters are simulated and optimized by HFSS to find the optimal parameters to realize the miniaturization of the sensor.

### 3.1. CPW Feed Line Index Asymptotic Linearization Design

In micro strip antennas, the purpose of reducing standing waves and improving transmission efficiency is often achieved by using micro strip impedance converters. In this paper, the exponential asymptotic linearization of the CPW feed line makes the CPW feed line guide band section divide into numerous steps from bottom to top, and makes the step length of each section infinitely short when the width and characteristic impedance of the CPW feed line change continuously, so that the reflections generated by each step cancel each other and realize the impedance matching of the antenna in the wide band. The micro strip characteristic impedance *Z_c_* of the CPW feed line varies along the impedance converter length direction, which varies exponentially according to the following calculation equation:(20)$$Z_{c}=Z_{02}e^{\alpha z}$$

In the formula, *Z*_02_ represents the micro strip characteristic impedance at the center of the asymptote, and α represents the transformation constant of the characteristic impedance.

The Txline software calculates the micro strip characteristic impedance *Z*_1_ and *Z*_2_ (*Z*_1_ > *Z*_2_) for matching the upper bottom width w_1_ and the lower bottom width w_2_ of the CPW feeder, which in turn calculates the length *l* of the guide strip by the following equation:(21)$$l=\frac{\lambda \ln(Z_{1}+Z_{2})}{8\pi \Gamma_{1}}$$

In the equation
(22)$$Z_{1}=Z_{02}e^{-\alpha l/2}$$
(23)$$Z_{2}=Z_{02}e^{\alpha l/2}$$

Γ_1_ denotes the voltage reflection coefficient of the exponentially asymptotic linearize impedance converter, which corresponds to the following calculation equation
(24)$$\Gamma_{1}=\frac{\lambda }{8\pi l}\ln\frac{Z_{1}}{Z_{2}}$$

By calculation, we can obtain *l* = 110 mm, so the ground plane size can be reduced, that is, the height H of the trapezoidal ground plane is also taken as close to 110 mm to match the CPW part. The antenna corresponding transverse length A is 180 mm and the longitudinal length B is 170 mm. The CPW feed line is exponentially asymptotically linearize and two new variables are derived from the feed line guide band upper bottom width w_1_ and lower bottom width w_2_, which are optimized by the next simulation to find the optimal parameters based on the original feed line width w = 2.5 mm.

### 3.2. Antenna Simulation Optimization Design

The size of the antenna is still large after the asymptotic linearization of the CPW feed index, and further simulation optimization is needed to find the optimal parameters of the reduced antenna. Because the main factor affecting the bandwidth of monopole antenna is the size of monopole antenna patch, according to the antenna structure, it is known that the two main parameters affecting the area of monopole antenna are A and B. The parameters a, b, H, H_1_, D_min_, D_max_, w_1_, w_2_, d_1_ only need to be changed accordingly with the change of parameters A and B. If the 11 parameters are optimized by traditional parameter scanning, the computational effort is very large. In order to ensure the implementability of the work, this paper first simulates and optimizes the parameters A and B to find the optimal values and derives the approximate ranges of the parameters a, b, H, etc. Then the fine optimization of the parameters a, b, H, etc. is processed to finally determine the optimal values of the 11 parameters.

Equation (25) represents the formula for the 1/4 wavelength corresponding to the electromagnetic wave in free space, where *f*_1_ indicates the preset operating frequency of the antenna.
(25)$$L=\frac{c}{4f_{1}\sqrt{\varepsilon_{re}}}$$

Parameter optimization domains of A and B are determined by Equation (25) for A: 120 to 170 mm and B: 120 to 180 mm.

According to the set optimization domain, the parameter A is first optimized by using ANSYSS HFSS software, and the simulation results obtained are shown in Figure 4. It can be seen from the figure that as the value of parameter A decreases, the center frequency of the low frequency band gradually moves to the right near the target center frequency and the bandwidth gradually increases. After the parameter A is reduced to 140 mm, the bandwidth appears to be degraded and narrowed, so A = 140 mm is determined.

After determining the value of parameter A, the parameter value of A was set to 140 mm, followed by the optimization of parameter B. The simulation results obtained are shown in Figure 5. It can be seen from the figure that as the value of parameter B decreases, the fluctuation of the center frequency gradually decreases and moves to the right near the target center frequency, and the bandwidth also increases with it. After the parameter B is reduced to 150 mm, bandwidth is degraded and narrowed, so B = 150 mm is determined.

After determining the values of parameters A and B, the obtained VSWR curve center frequency coincides with the target center frequency; the bandwidth approaches the limit width as the target; the parameters a, b, H, H_1_, D_min_, D_max_, w_1_, w_2_, d_1_ are fine-tuned by simulation optimization; and the optimal VSWR curve obtained by optimization is shown in Figure 6.

The final structure of the antenna determined according to the final VSWR curve is shown in Figure 7. The corresponding specific structural parameters are shown in Table 1, where the cone angle *θ*_0_ of the trapezoidal ground plane is taken as 32°.

### 3.3. Flexible Substrates

To address the shortcomings of the internal UHF sensor using rigid materials as the substrate proposed in the previous paper, this paper uses flexible materials as the substrate of the antenna. Common flexible materials are PI (polyether imide) [23], PDMS (polydimethylsiloxane) [24], PTFE (poly tetra fluoroethylene) [25], etc. Table 2 gives the performance parameters of PI, PDMS, and PTFE flexible materials. The performance parameters of PI, PDMS, and PTFE are given in Table 2. The flexible material PI used in this paper has the advantages of high dielectric constant and low dielectric loss, which can ensure the high efficiency of the antenna during signal transmission. In addition, due to its good insulation, ductility and flexibility, it can guarantee the flexibility and stability of the flexible antenna sensor in the process of internal GIS.

In this paper, the designed antenna is printed on a rectangular PI flexible dielectric plate, the length of which is 150 mm, the width of which is 140 mm, and the thickness of which is 0.28 mm. The SMA-KE interface of the RF connector is used to connect the SMA CPW feeder for feeding, The physical diagram of the antenna is shown in Figure 8.

### 3.4. Antenna Performance Testing

#### 3.4.1. Voltage VSWR

The voltage standing wave ratio (Voltage Standing Wave Ratio, VSWR) indicates the ratio of the wave web voltage and the wave node voltage, usually with the standing wave ratio being lower than a certain value of the bandwidth definition of the antenna impedance bandwidth. This can reflect both the frequency characteristics of the antenna impedance, but also the matching effect between the antenna and the feed line; generally the VSWR bandwidth is less than 2, which is the effective frequency band of the UHF sensor. VSWR calculation formula is as follows.
(26)$$\mathrm{VSWR}=\frac{1+\left|\Gamma \right|}{1-\left|\Gamma \right|}$$
where Γ is the reflection coefficient of the antenna.

The GIS shell in a power system is a cartridge structure, and its shell bending radius is generally between 150~500 mm depending on the voltage level and manufacturing process. In this paper, the three-dimensional electromagnetic simulation model of elliptical monopole antenna is established by ANSYS HFSS software 15.0, and the VSWR curves are shown in Figure 9 when the flexible antenna is not bent (0 mm) and when the bending radius is 100, 300, and 500 mm, respectively, in the frequency band of 300 MHz~3 GHz. The simulation results show that when the flexible antenna designed in this paper is not deformed, VSWR < 2 in the frequency band of 340 MHz to 3 GHz. After different degrees of bending and deformation, the antenna VSWR will have small fluctuations, but the fluctuations are very small: when the bending radius is 100 and 300 mm, the antenna VSWR in the 340 MHz–3 GHz band is <2; when the bending radius is 500 mm, the antenna VSWR in the 500 MHz–750 MHz band is <2.5; and in the 750 MHz~3 GHz band, the VSWR < 2.

Furthermore, using Agilent Vector Network Analyzer (E5063A), the physical VSWR of the antenna was measured, and the test results are shown in Figure 10. It can be seen from the figure that when the bending radius of the antenna designed in this paper is 0, 100, 300, and 500 mm, in addition to the VSWR < 5 in the 300 MHz~650 MHz frequency band, the VSWR of the 650 MHz~3 GHz frequency band is basically <2, indicating that the VSWR is basically unchanged after the antenna bending deformation.

Comparing the simulated and measured antenna VSWR results, there is a certain difference between the measured VSWR in the low frequency band and the simulated results, which is due to the influence of the antenna production and welding process precision and the unavoidable metal conductor interference in the test environment, resulting in the narrowing of the measured VSWR bandwidth in the low frequency band and oscillation in the high frequency band. However, the overall effect meets the antenna design requirements.

#### 3.4.2. Radiation Direction Map

The radiation pattern indicates that in the far field of the antenna, the relative field strength of the radiated field changes with the direction, which is used to feed back the gain effect of the antenna. The flexible antenna designed in this paper (bending radius of 0, 100, 300, and 500 mm) at 0.5, 1, 1.5, 2, 2.5, and 3 GHz six frequency points and the radiation pattern of the antenna plane H (YOZ plane) and plane E (XOY plane) are shown in Figure 11 and Figure 12.

From Figure 11 and Figure 12, it can be seen that the H-plane directional pattern is inverted “8” at 0.5, 1, 1.5, 2, and 3 GHz 5 frequency points, and the E-plane directional pattern is inverted “8” at 0.5, 1, 1.5, and 3 GHz 4 frequency points. Except for some frequency points, it can receive UHF signal better, and the antenna can receive UHF signal better in H-plane. At the same time, it can be seen from the directional diagram that under the same bending radius, the receiving signal effect of the H-plane of the flexible antenna increases with the increasing of frequency; under the same frequency, the radiation characteristics of the H-plane of the flexible antenna are less affected by the curved bending deformation. This is because the main body of the flexible antenna is monopole antenna. The monopole antenna designed in this paper has a mainly symmetrical structure along the H-plane, so its main received signal direction is in the H-plane, which leads to the deterioration of the antenna E-plane gain. Since the antenna itself is installed with its H-plane facing the UHF signal transmitting direction, it does not affect the perception performance of the antenna itself.

## 4. Flexible Antenna Built-In Test

When the flexible antenna is built into GIS, it may affect its internal electric field distribution and become a potential risk source, so the impact of the flexible antenna on the internal electric field distribution of GIS must be taken into account after it is built into GIS. In this paper, a simulation model of GIS internal flexible antenna is built by COMSOL Multiphysics to analyze the effect of internal antenna on the internal electric field distribution of GIS. The flexible antenna is installed on the inner wall of GIS metal shell, 30 kV is applied to the GIS high-voltage guide rod (bending radius 40 mm), and the GIS (bending radius 400 mm) is grounded; the antenna adopts a flexible substrate (thickness 0.28 mm) and the surface copper layer (thickness 0.035 mm) to form a simplified model, as shown in Figure 13.

Simulations were carried out for the two cases where the flexible antenna was not built in and the flexible antenna was built in within the range of 0~400 mm from the high-voltage conduit, so as to better analyze the influence of the built-in flexible antenna on the electric field distribution inside the GIS. The simulation results are shown in Figure 14.

From the simulation results, it can be seen that the internal field strength of GIS with internal antenna is basically the same as the internal field strength without internal antenna. The internal electric field distribution law of GIS is not destroyed because of internal flexible antenna, so the effect of flexible antenna internal GIS on the internal electric field distribution of GIS is almost negligible.

## 5. Flexible Antenna Performance Testing and Verification

### 5.1. PD Testing Test Platform

According to the test requirements of the discharge capacity in the partial discharge test of the power equipment, the International Electrotechnical Commission (IEC) has formulated a special standard for this method: IEC60270 [26]. China has also formulated a corresponding testing standard on this basis: GB/T7354-2018 [27]. This method usually measures the electrical pulse signal generated by the device PD by connecting the coupling device (RL or RLC circuit) to the test sample in series or to the coupling capacitor, as shown in Figure 15, where U is the high-voltage power supply, C_a_ is the test product, C_k_ is the coupling capacitor, CD is the coupling device, and M is the signal detection device.

To verify the performance of the flexible antenna designed in this paper to detect PD signals, aiming at the two standards of IEC60270 and GB/T7354-2018, this paper builds a GIS partial discharge test circuit as showed in Figure 16, and uses the pulse current method to obtain the magnitude of partial discharge. Among them, 7 corresponds to M in Figure 15a, 5 corresponds to Ck in Figure 15a, and 6 corresponds to CD in Figure 15a. By filling the real type 220 kV GIS cavity with 0.5 MPa SF_6_ gas and setting a typical metal filth discharge defect in it. The developed flexible UHF antenna sensor is bent and affixed into the cavity wall for PD radiation electromagnetic pulse sensing performance testing. The circuit of the test platform consists of an industrial frequency power supply, a transformer without partial discharge, a protection resistor, a voltage dividing capacitor, a coupling capacitor, a 220 kV GIS, etc. The signal acquisition equipment uses a Tektronix high- performance digital oscilloscope (Tektronix*MSO44, four channels, bandwidth 1.5, sampling frequency 6.25 GS/s) and the metal filth discharge defect in kind is shown in Figure 17.

When the test voltage is 12.5 kV and the average discharge is about 9.5 pc, the flexible antenna sensor designed in this paper can effectively detect the high-frequency electromagnetic wave signal radiated under this discharge intensity. Figure 18 shows the time domain (Figure 18a) and frequency domain characteristics (Figure 18b) of a UHF electromagnetic pulse signal detected at a sampling rate of 6.25 Gs/s.

From Figure 18, it can be seen that the main frequency range of UHF electromagnetic pulse is from 1 GHz to 1.5 GHz, and the main interfering frequency points of electromagnetic pulse noise in the band range of 300 MHz–3 GHz are 376.42 MHz, 878.50 Hz, 938.75 MHz, and 1.86 GHz. Analyzing the frequency points, 376.42 MHz, 878.50 MHz, 938.75 MHz, and 1.86 GHz interference points are caused by 4 G signals from nearby base stations. In the UHF spectrum, the electromagnetic interference frequency component accounts for a small percentage. After excluding the influence of external electromagnetic waves, the flexible antenna can effectively detect the UHF electromagnetic wave signals from 1 GHz to 1.5 GHz radiated by metal fifth metal discharge defect, and the designed flexible antenna can effectively realize the detection of UHF electromagnetic wave signals radiated by metal fouling defects with low discharge amount and excellent antenna performance.

### 5.2. PD Signal Detection Test under Different Voltage Levels

To verify the PD detection performance of the flexible antenna after bending and deformation, the antenna was bent and placed inside the GIS cavity surface. The test voltage was stepped from 12.5 kV to 13.6 kV, the PD performance detection test was performed on the flexible antenna, and the PD signal waveform was collected at the sampling rate of 6.25 GS/s, as shown in Figure 19.

From Figure 19, it can be seen that the amplitude of the signal collected by the flexible antenna is 87.2 mV when the voltage level is 12.5 kV, and 88.0 mV when the voltage level is 13.6 kV at a sampling rate of 6.25 GS/s. The test shows that the flexible antenna has good performance in detecting PD signals at different voltage levels.

### 5.3. PD Signal under the Average Discharge of the Average Pulse Amplitude Detection Test

Because partial discharge itself has a certain fluctuation, in order to further test the sensing performance of the flexible antenna sensor designed in this paper, the spectrum information of PD pulse current signal and UHF electromagnetic pulse signal sensed by flexible antenna at sampling rate of 5 MS/s were analyzed under 12.5 kV test voltage level, and the statistical results are shown in Figure 20 and Figure 21.

From Figure 20 and Figure 21, it can be seen that both PD pulse current signal and UHF electromagnetic pulse signal are concentrated in the positive half-period peak and negative half-axis peak of AC phase, and the average amplitude of PD pulse current signal is 6.71 mV, corresponding to the average discharge amount of 9.62 pC; the average amplitude of UHF electromagnetic pulse signal is 5.45 mV (at 5 MS/s sampling rate), which indicates that the antenna designed in this paper can effectively detect the discharge signal below 10 pC with high sensitivity.

## 6. Performance Comparison

In order to further verify the performance effects of the antenna designed in this paper, such as gain and sensitivity, this paper selects three articles about the built-in UHF PD detection antenna for comparison [12,13,28], and compares the performance of size, gain, sensitivity, etc. The performance comparison is shown in Table 3.

From the comparison results in the Table 3, on the basis of being built in, the antenna designed in this paper has flexibility, and the bending deformation has almost no effect on the detection of PD signals by the antenna; high gain at multiple frequencies of 0.5, 1, 1.5, 2.5, and3 GHz; it can detect PD signals below 10 Pc, and capture the effect of UHF signals, with high detection sensitivity.

## 7. Conclusions

To address the shortcomings of the existing built-in UHF sensor with rigid substrate, this paper carried out the study of flexible built-in high-sensitivity elliptical monopole antenna sensor for PD detection, introduced PI material as the flexible base of the antenna, and simulated and verified the emission characteristics of the antenna through finite element simulation and analyzer real measurement. Finally, the PD signal detection performance of the antenna was measured empirically using the constructed experimental platform, and the following conclusions were drawn.

Three techniques, rectangular patch antenna equivalent technique, trapezoidal ground plane technique, and CPW feed line index asymptotic linearization were used to reduce the VSWR and expand the bandwidth of the antenna in the 300 MHz~3 GHz band. The simulated and measured results show that a VSWR < 3 can be maintained in the 300 MHz~650 MHz bands under the bending radius of 0, 100, 300, and 500 mm and VSWR < 2 can be maintained in the 650 MHz~3 GHz bands, with good omnidirectional radiation characteristics.The flexible antenna was tested by software simulation and GIS PD testing platform, which showed that the influence of GIS on the internal electric field distribution of the flexible antenna was almost negligible, which was suitable for practical application.The flexible antenna was tested by the built GIS PD detection test platform, and the test showed that the flexible antenna can effectively detect the PD signal with an average discharge below 10 pC after bending, and the sensitivity of the test PD signal is high.

## Figures and Tables

**Figure 1 sensors-22-04982-f001:**
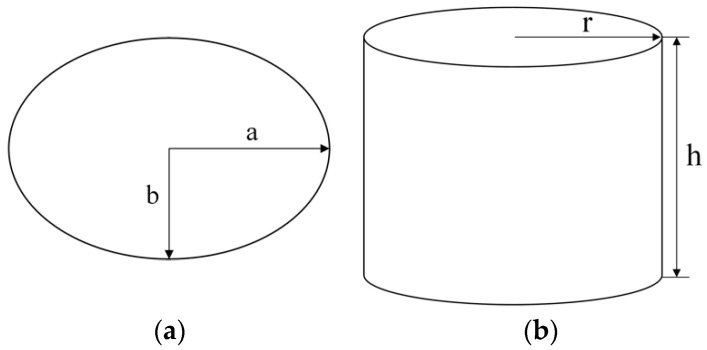
Calculation principle of elliptical radiation patch: (**a**) elliptical radiation patch principle; (**b**) cylindrical oscillator.

**Figure 2 sensors-22-04982-f002:**
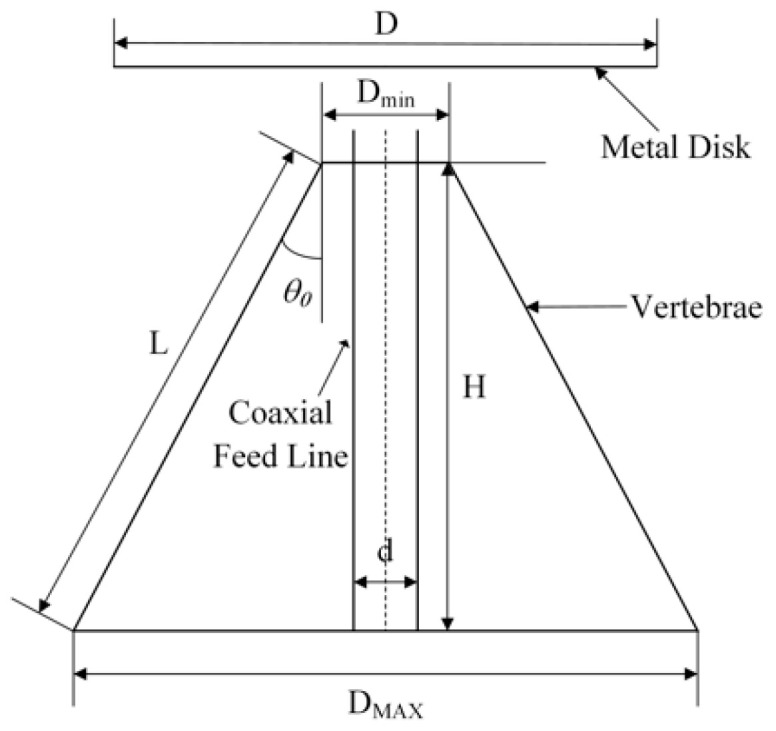
Schematic diagram of discone antenna.

**Figure 3 sensors-22-04982-f003:**
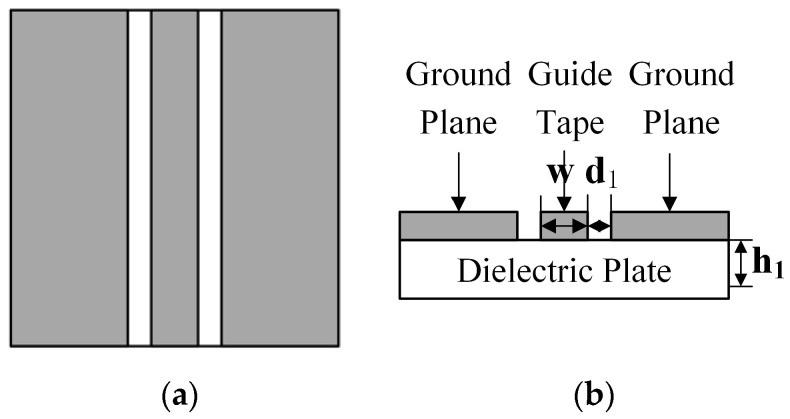
Schematic diagram of CPW feed line: (**a**) top view of CPW feed line; (**b**) section diagram of CPW feed line.

**Figure 4 sensors-22-04982-f004:**
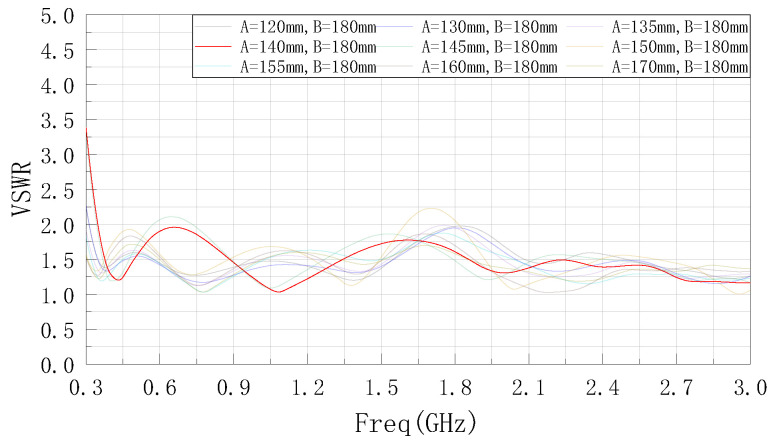
VSWR curve after optimized parameter A.

**Figure 5 sensors-22-04982-f005:**
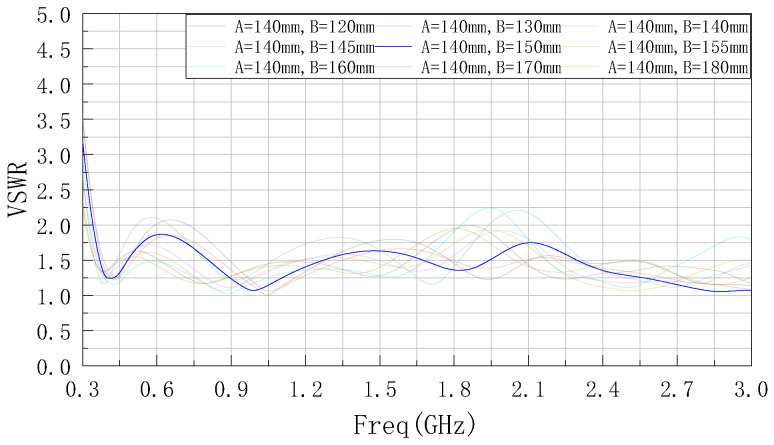
VSWR curve after optimized parameter B.

**Figure 6 sensors-22-04982-f006:**
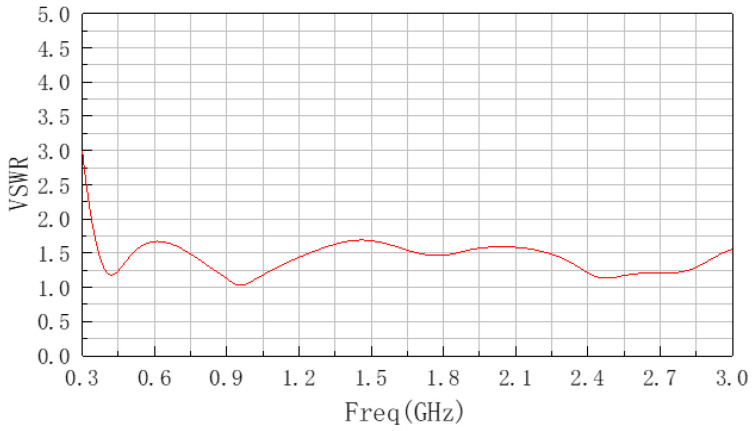
The optimal VSWR curve.

**Figure 7 sensors-22-04982-f007:**
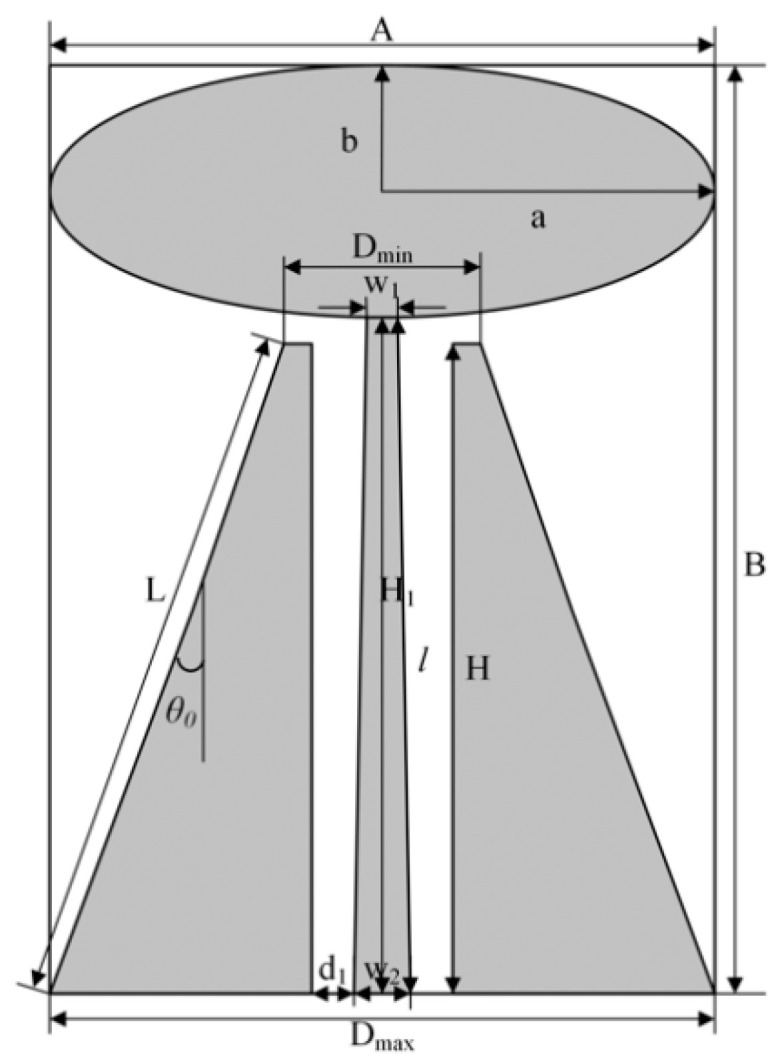
Structure diagram of elliptic monopole antenna.

**Figure 8 sensors-22-04982-f008:**
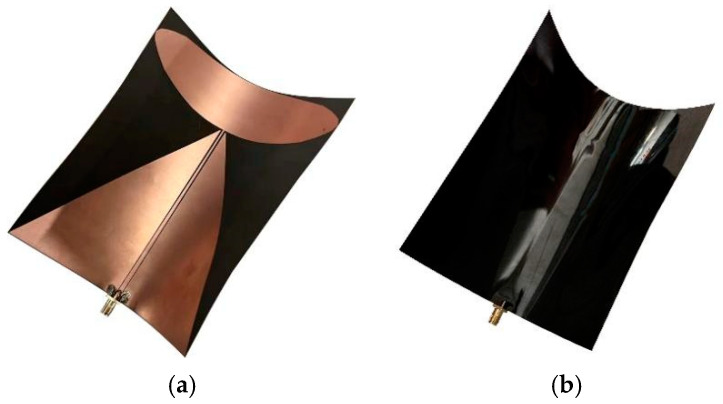
Physical view of the antenna: (**a**) front side of the antenna; (**b**) back side of the antenna.

**Figure 9 sensors-22-04982-f009:**
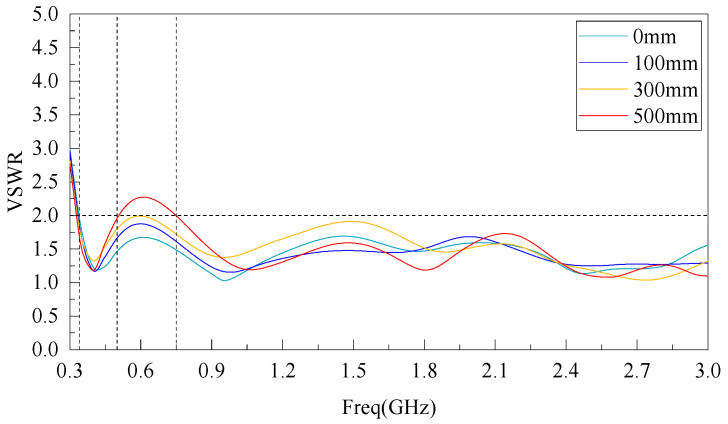
Simulation of VSWR curve.

**Figure 10 sensors-22-04982-f010:**
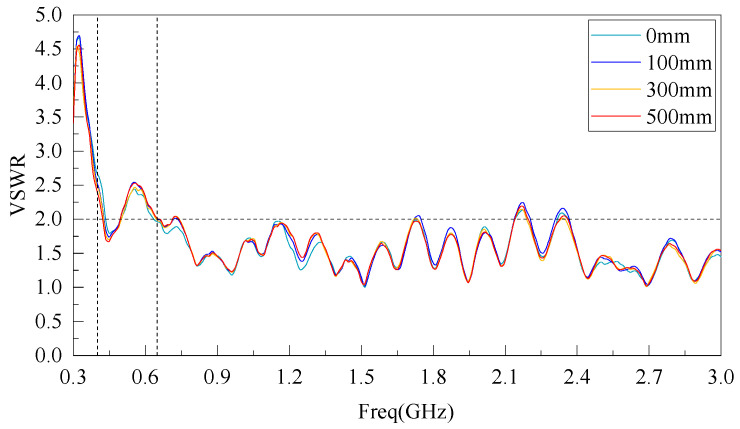
Measured VSWR curve.

**Figure 11 sensors-22-04982-f011:**
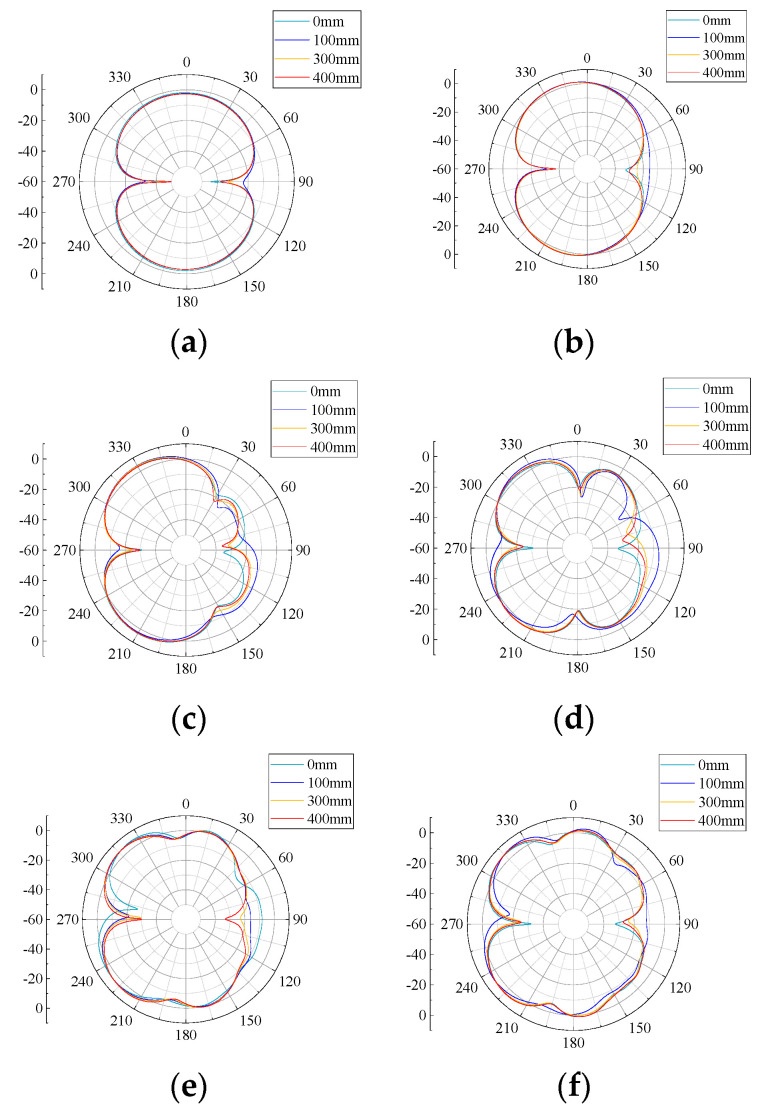
H-plane patterns at different frequencies: (**a**) 0.5 GHz; (**b**) 1 GHz; (**c**) 1.5 GHz; (**d**) 2 GHz; (**e**) 2.5 GHz; and (**f**) 3 GHz.

**Figure 12 sensors-22-04982-f012:**
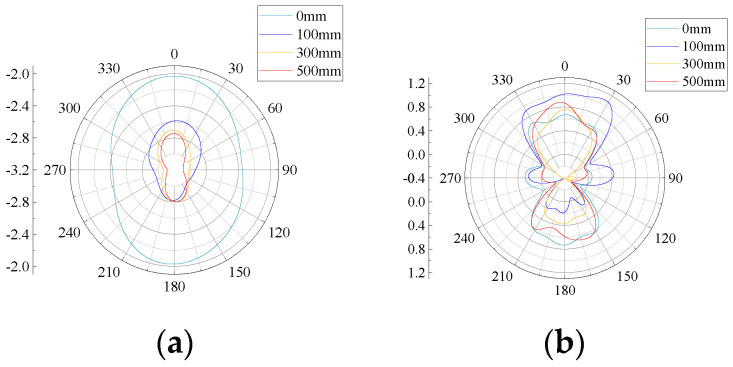
E-plane patterns at different frequencies: (**a**) 0.5 GHz; (**b**) 1 GHz; (**c**) 1.5 GHz; (**d**) 2 GHz; (**e**) 2.5 GHz; and (**f**) 3 GHz.

**Figure 13 sensors-22-04982-f013:**
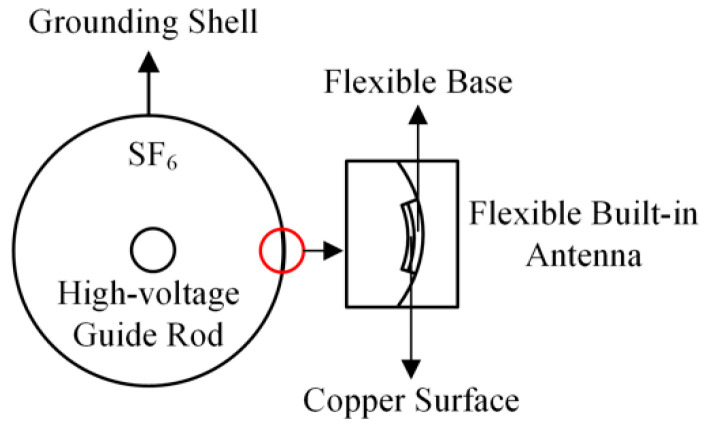
GIS built-in simplified model.

**Figure 14 sensors-22-04982-f014:**
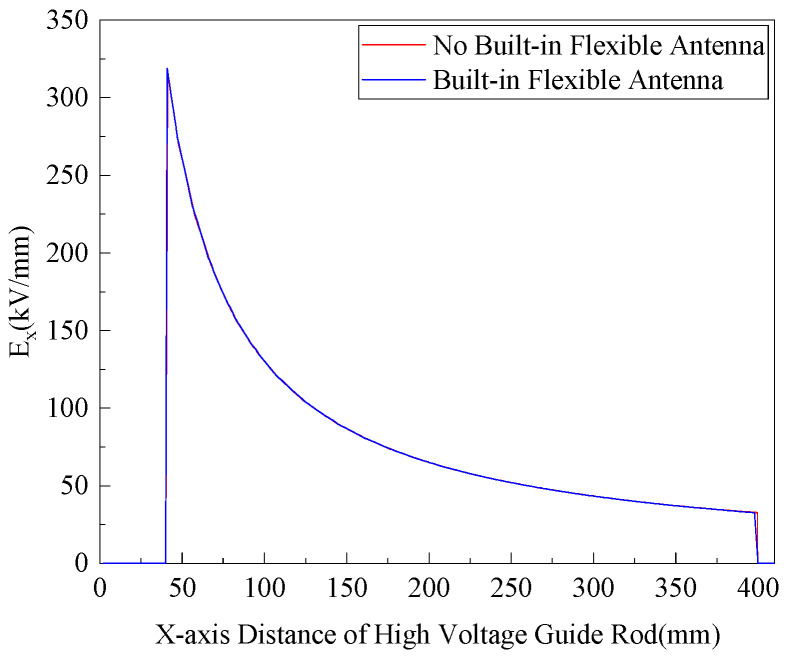
Field strength change in the *x*-axis 0~400 mm.

**Figure 15 sensors-22-04982-f015:**
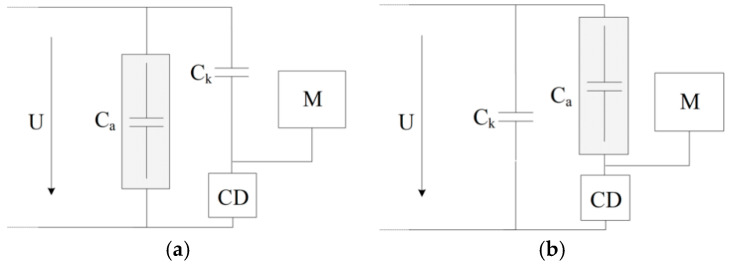
Typical pulse current method PD measurement circuit: (**a**) the coupling device is connected in series with the coupling capacitor; (**b**) the coupling device is connected in series with the test sample.

**Figure 16 sensors-22-04982-f016:**
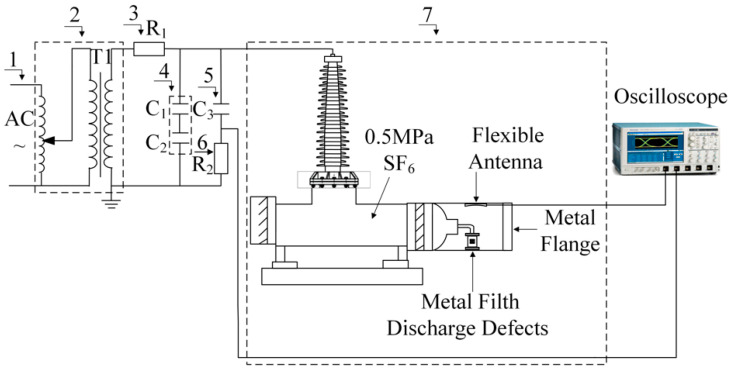
Experimental circuit 1-Industrial frequency power supply; 2-No partial discharge transformer; 3-Protection resistor; 4-Dividing capacitor; 5-Coupling capacitor; 6-Detection impedance; 7–220 kV GIS.

**Figure 17 sensors-22-04982-f017:**
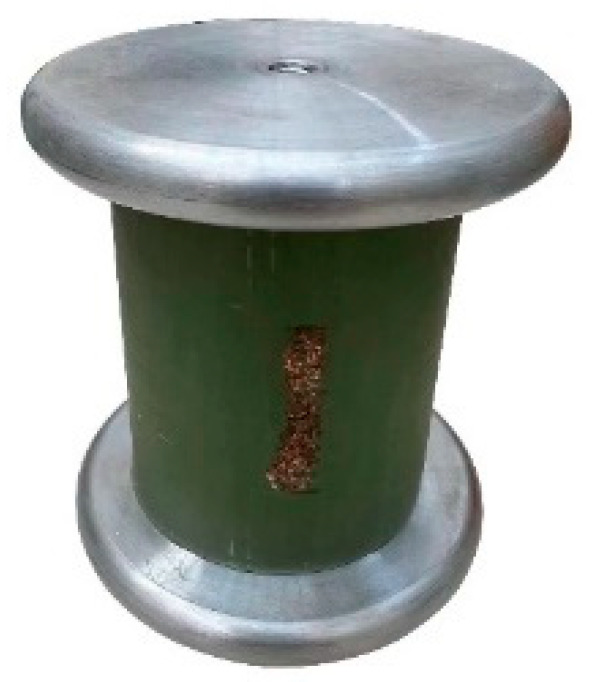
Metal filth discharge defect.

**Figure 18 sensors-22-04982-f018:**
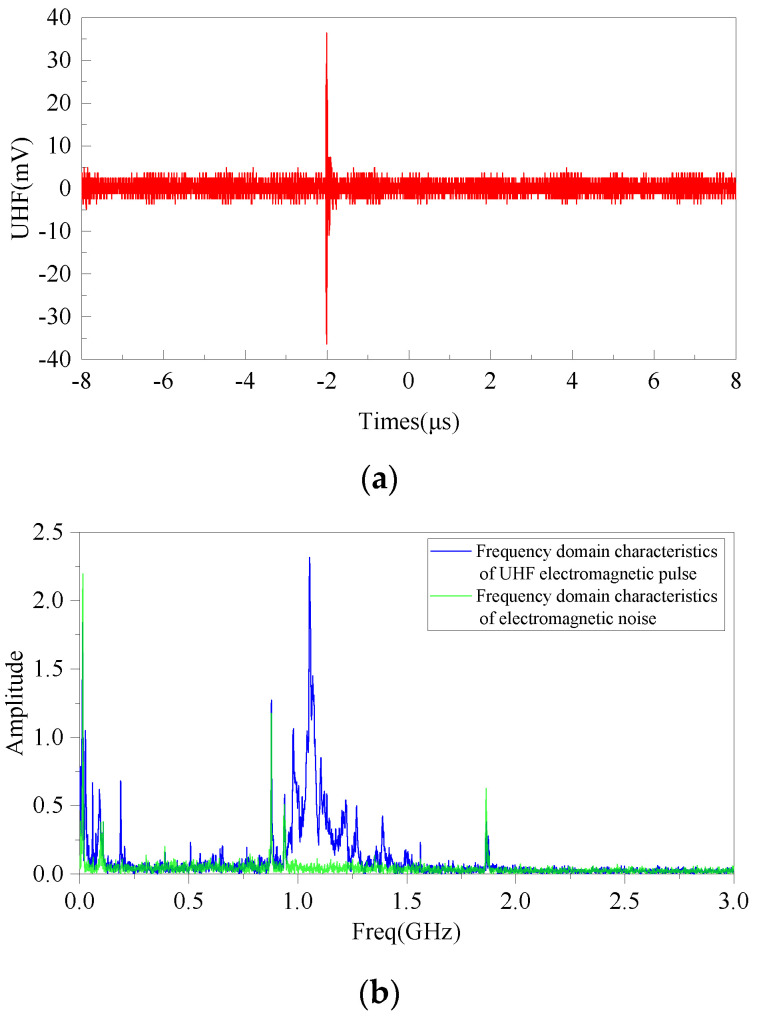
Metal filth discharge defects: (**a**) time domain characteristics; (**b**) frequency domain characteristics.

**Figure 19 sensors-22-04982-f019:**
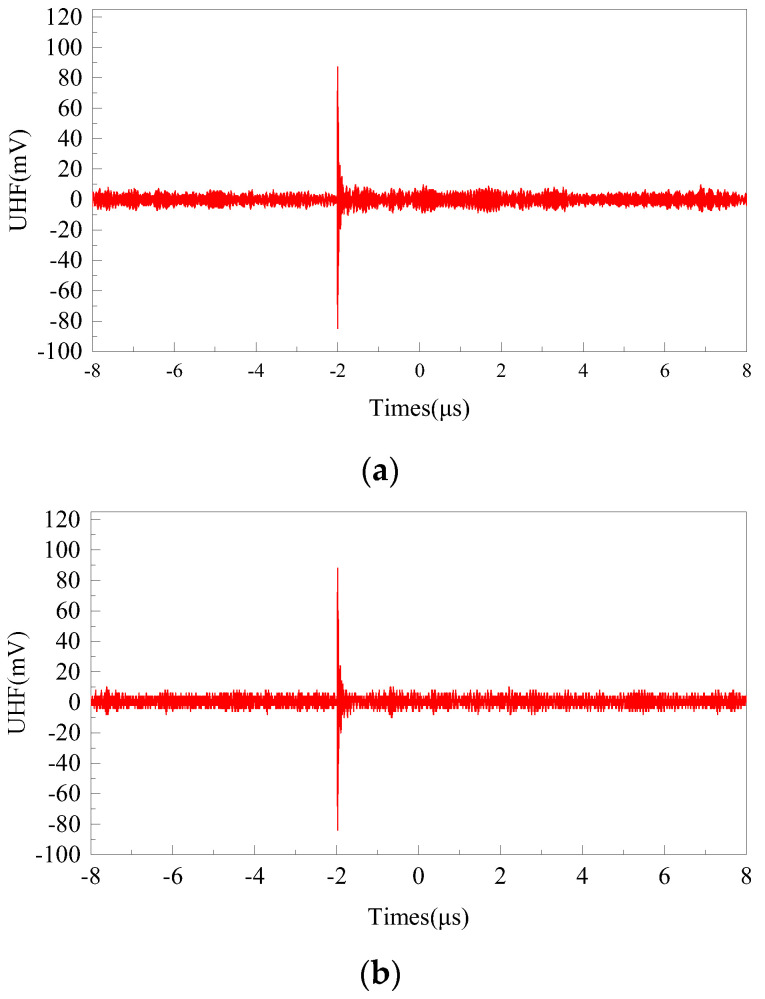
Signal waveform of flexible antenna collected at two voltage levels: (**a**) 12.5 kV and (**b**) 13.6 kV.

**Figure 20 sensors-22-04982-f020:**
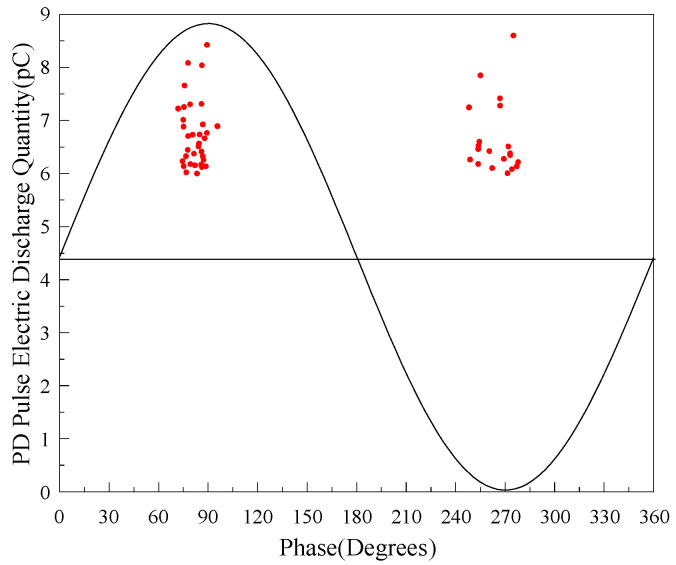
Discharge pattern of the filth defects inside GIS under 12.5 kV.

**Figure 21 sensors-22-04982-f021:**
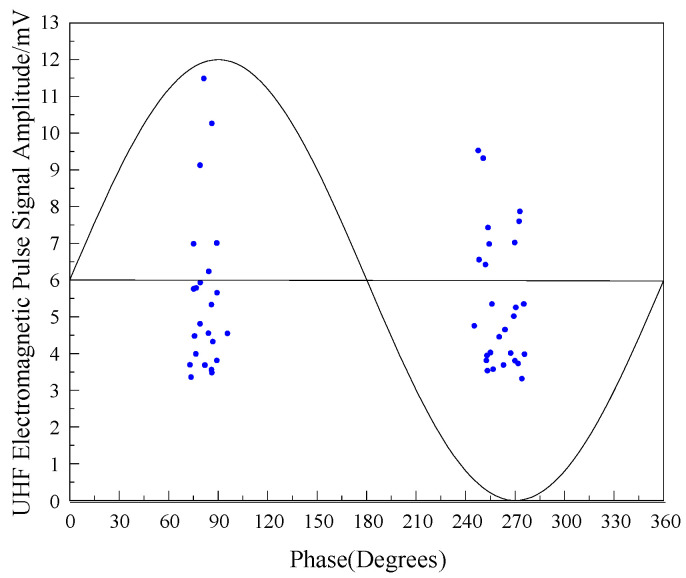
UHF electromagnetic pulse signal amplitude pattern of the filth defects inside GIS under 12.5 kV.

**Table 1 sensors-22-04982-t001:** Design parameters of Antenna.

Parameter		Numerical Value/mm
Base	Transverse length A	150
	Longitudinal length B	140
	Thickness h_1_	0.28
Elliptical radiation patch	Long axis a	70
	Short axis b	17.5
Trapezoidal ground plane	Bus length L	133
	Height H	114
	Upper base D_min_	4
	Lower base D_max_	140
CPW feed line	Guide tape upper bottom width w_1_	0.87
	Guide tape lower bottom width w_2_	3
	Distance between guide tape and ground plane d_1_	0.18
	Guide tape hypotenuse length *l*	115
	Guide tape band height H_1_	115

**Table 2 sensors-22-04982-t002:** Flexible material performance parameters.

	Materials	PI	PDMS	PTFE
Parameters				
Dielectric constant		3.5	2.7	2.2
Dielectric loss		0.004~0.007	0.001~0.004	0.001~0.005
Breakdown field strength kv/mm		300	20	200

**Table 3 sensors-22-04982-t003:** Performance comparison.

	Literature	This Article	Literature [13]	Literature [12]	Literature [28]
Performance					
Size/mm		140 × 150	195 × 142	124 × 77	145 × 145
Is it flexible		Yes	Yes	No	Yes
VSWR		340 MHz~3 GHz < 2	450 MHz~3 GHz < 5	500 MHz~3 GHz < 5	500 MHz~3 GHz < 5
Does the deformation have a huge impact on VSWR		No	No	None	No
Inverted “8” radiation pattern frequency points/GHz		0.5,1,1.5,2.5,3	0.5,0.7,1,1.5	0.66,1.5,2.5	0.6,1,1.5,2,2.5,3
The smallest detectable discharge size/Pc		8.6	13.9	None	18.6
UHF signal detection spectrum analysis		The effect of capturing UHF signal is obvious	The effect of capturing UHF signal is poor	The effect of capturing UHF signal is obvious	The effect of capturing UHF signal is obvious

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
