# Peer review of "PD Flexible Built-In High-Sensitivity Elliptical Monopole Antenna Sensor"

_sensors, 2022, doi:10.3390/s22134982_

Round 1

Reviewer 1 Report

The authors present a PD Flexible Built-in UWB High-Sensitive Elliptical Monopole Antenna Sensor. The topic is interesting, PD monitoring using radiometric methods is a realy hot topic nowadays.
Despite the promising results presented, in my opinion, this paper needs improvement.
1 - Firstly, careful proofreading is mandatory.
2 - In the introduction, more information is expected, the problem to solve is not clear, more references are requested in order to situate the structure proposed. There are a lot of references in the literature about monopole antennas for PD monitoring, for GIS and power transformers applications, I can cite the bioinspired monopole antennas which have already been published in this journal.
3 - Regarding the theoretical background, Elliptical plane monopole antenna is a well known concept and can be resumed. I have the same comments for the Trapezoidal ground plane and the CPW feed line.
4 - Throughout the paper "return loss" must be replaced by "reflection coefficient",  the figures show S11 and not RL with negative values.
5 - All parameters of the antenna are requested in Table 1.
6 - Figure 8 must be improved.
7 - I don't understand section 3.4.1. Voltage VSWR, all the simulations were performed showing the S11 and for measurements the authors used the VSWR, moreover the Equ.26 needs to be checked.
8 - In the section Radiation direction map, realized gain values are requested with measurement comparison.
9 - PD signal detection with the antenna proposed must be compared to the IEC 62271-203 method.
10 - Finally, the contribution must be compared to other designs found in the literature (size, gain, sensibility...)

Author Response

Thanks for your comments and questions! Your comments and questions are expected to be very valuable! In response to your comments and questions, we have carefully revised the paper, please see the attachment.

Reviewer 2 Report

The authors presented a PD Flexible Built-in UWB High-Sensitive Elliptical Monopole Antenna Sensor.

The concept is exciting, and the simulation results are reasonably good, showing potentially strong reconfigurability. However, this article missed some of the essential principles, as follows:

Title:

The title is incorrect for me; let me explain this, the Elliptical Monopole antenna operated at frequencies from 0.3 GHz to 3 GHz. These frequencies are considered a wide band operation, not an Ultrawideband (UWB). The Ultrawide-band frequencies must cover the FCC licenced band from 3.1 GHz to 10.6 GHz. We cant only say ultrawideband without having the evidence. Please recheck ?!

Abstract

  • What is PD?
  • What GIS?
  • What is UHF?
  • What is CPW?
  • What is VSWR?

The authors should propose the Acronyms above before using them.

-What is the Elliptical Monopole antenna gain??

Introduction

The introduction is insufficient and needs improvement. A total of 10 references in the introduction part are not enough. Please discuss some UWB planar (Monopole) antennas and their ability to enhance the gain performance [1,2]. Please see these articles, which may add value to the introduction [1,2].

[1] Enhancing Gain for UWB Antennas Using FSS: A Systematic Review. Mathematics2021; 9(24):3301. https://doi.org/10.3390/math9243301.

[2] Ultra-Wideband Antennas for Biomedical Imaging Applications: A Survey. Sensors 202222, 3230. https://doi.org/10.3390/s22093230.

3.3. Flexible substrates

-Please propose the simulated and measured reflection coefficient (S11) or the simulated and measured VSWR of the suggested planar antenna??

-Please present the gain graph of the proposed antenna?

-It would be nice if the authors compared the existing literature and listed it in a Table.

That's all for me at this moment! The authors are required to revise the comments above carefully. Thanks

Author Response

(The authors gave the same response as above.)

Round 2

Reviewer 1 Report

All my comments were attended. The paper was improved and in my opinion suitable for publication in this form.

Author Response

Thank you very much for your recognition of our paper! All the authors of the paper have carefully corrected the paper again to make the paper more perfect.

Reviewer 2 Report

The authors have revised the given comments successfully. However, there are still typos and spacing errors that need to be carefully cehck.

Author Response

Thanks a lot for your opinion! According to your opinion, all the authors of the paper have carefully corrected the paper again, corrected the spelling errors and spacing errors, and made the paper more perfect.